# Implementing Divisive Normalization in CNNs Improves Robustness to Common Image Corruptions

**Andrew Cirincione**[1,*], **Reginald Verrier**[1,*], **Artiom Bic**[1,*], **Stephanie Olaiya**[2],
**James J. DiCarlo**[3,4,5], **Lawrence Udeigwe**[1,3,†], **Tiago Marques**[3,4,5,6,†,§]

[1]Department of Mathematics, Manhattan College, Riverdale, NY10471
[2]Department of Neuroscience, Brown University
[3]Department of Brain and Cognitive Sciences, MIT, Cambridge, MA02139
[4]McGovern Institute for Brain Research, MIT, Cambridge, MA02139
[5]Center for Brains, Minds and Machines, MIT, Cambridge, MA02139
[6]Champalimaud Clinical Centre, Champalimaud Foundation, Lisbon

[*] Joint first authors (equal contribution)
[†] Joint senior authors (equal contribution)
[§] Corresponding author: `tiago.marques@research.fchampalimaud.org`

## Abstract

Some convolutional neural networks (CNNs) have achieved state-of-the-art performance in object classification. However, they often fail to generalize to images perturbed with different types of common corruptions, impairing their deployment in real-world scenarios. Recent studies have shown that more closely mimicking biological vision in early areas such as the primary visual cortex (V1) can lead to some improvements in robustness. Here, we extended this approach and introduced at the V1 stage of a biologically-inspired CNN a layer inspired by the neuroscientific model of divisive normalization, which has been widely used to model activity in early vision. This new model family, the VOneNetDN, when compared to the standard base model maintained clean accuracy (relative accuracy of 99%) while greatly improving its robustness to common image corruptions (relative gain of 18%). The VOneNetDN showed a better alignment to primate V1 for some (contrast and surround modulation) but not all response properties when compared to the model without divisive normalization. These results serve as further evidence that neuroscience can still contribute to progress in computer vision.

## 1 Introduction

Over the past decade, some convolutional neural networks (CNNs) have achieved state-of-the-art performance in several computer vision applications such as image classification [1, 2, 3]. In parallel, CNNs have also been relatively successful in modeling neuronal responses along the primate ventral visual pathway as well as primate object recognition behavior, when compared to more traditional neuroscience models [4, 5, 6, 7, 8]. However, most CNNs exhibit important limitations such as their vulnerability to adversarial attacks [9, 10, 11], and, perhaps more relevant for real-world applications, failure to recognize objects in images perturbed with different types of common corruptions [12, 13, 14]. Additionally, no CNN tested to date is able to perfectly model the primate ventral visual stream to the extent that it has been tested [15, 7, 16].

4th Workshop on Shared Visual Representations in Human and Machine Visual Intelligence (SVRHM) at the Neural Information Processing Systems (NeurIPS) conference 2022. New Orleans.

To overcome some of these limitations, there have been several efforts to develop CNN models that are more directly inspired and constrained by neuroscience data [17, 18, 19, 20, 21, 22]. In [18], the authors developed a hybrid CNN containing a neuroscientific model of the primate primary visual cortex (V1) as its front-end, followed by a standard architecture back-end. This model family, the VOneNet, not only achieved high neural predictivity of V1 responses but also showed gains in robustness to adversarial attacks and common corruptions.

In this work, we expanded the VOneNet model family with a new layer, the DNBlock, inspired by the divisive normalization model, a popular neuroscientific model that has been used to explain a wide range of neuronal phenomena in primate early vision, such as surround suppression, cross-orientation inhibition, and nonlinear contrast response characteristics [23]. While this layer can be implemented in any CNN and in any of its stages, we introduced it at the output of the V1 stage of a VOneNet and called this model family, VOneNetDN. One key aspect of our implementation of the divisive normalization model was using an architecture and kernel parameterization (2D Gaussian) inspired in neuroscience, while optimizing its parameters for task performance using a standard deep learning approach. After optimizing the model parameters for object classification performance, including those in the divisive normalization layer, the new model shows not only a better clean accuracy, but also an improved robustness to common image corruptions, particularly the contrast and fog types, when compared to the VOneNet. In addition, it also better matches primate V1 at the single neuron level, with neurons exhibiting nonlinear contrast response function as well as surround modulation. Together, these results suggest that building models that more precisely emulate brain mechanisms can lead to further gains in challenging computer vision applications.

## 1.1 Related work

**Common corruptions:** Robustness of CNNs against common image corruptions [14] is an important topic that has received a great deal of attention lately. In [24], DeepAugment+AugMix, was considered the current state-of-the-art in this area. Augmentation with Gaussian noise and other noise patterns has also been shown to improve robustness [25, 26], but can affect performance on clean image [25] and low frequency corruptions [27]. Other approaches to increase robustness involve using: anti-aliasing module to restore the shift-equivariance [28], stylized images to increase shape bias [29, 25], stability training [30], and assembling several techniques [31].

**Biologically-constrained CNNs and robustness:** By replacing the first layer of a CNN with a model of primate V1, [18] was able to improve the model robustness to white box adversarial attacks and common corruptions. In a related work, [17] replaced the first convolutional layer of a standard CNN with Gabor filters to improve robustness to noise. A similar but complementary approach has been using neuronal representations to regularize training of models in object classification, which has also been shown to lead to better generalization [20, 21, 22].

**Divisive normalization:** Divisive normalization has been used to model a wide range of neuronal phenomena, namely in low-level visual areas [23]. In [32], a CNN with divisive normalization improved neuronal predictivity in the macaque V1. Brain-inspired divisive normalization has also been shown to lead to gains in image classification accuracy in small datasets [33].

## 2   Methods

We developed a novel model family containing a biologically-inspired divisive normalization layer at the model's V1 stage. We used the VOneNet [18], a family of CNNs whose front-end – the VOneBlock – simulates primate V1, as the starting point for our model. We adapted the VOneResNet18 variant with 512 channels in the VOneBlock (256 simple cells and 256 complex cells) and no neuronal stochasticity. The output of each VOneBlock channel is the result of convolving the input image with a biologically-constrained Gabor filter followed by either a simple- or complex-cell like nonlinearity (Figure 1). Then, we incorporated a divisive normalization layer, the DNBlock, that normalizes the activations of each VOneBlock channel by spatially pooling the activations of neighboring neurons in all the channels (weighted sum of the activation of VOneBlock channels). The normalizing spatial filters were constrained to have the shape of 2D Gaussians whose parameters are learned during training in object classification using the Tiny ImageNet dataset [34] (Section A.4 for more details). Parameterizing the normalization spatial filters with 2D Gaussians significantly reduces the number

of parameters to be optimized while allowing the divisive normalization model to capture lateral influence spanning 2deg of visual space. We call the resulting model the VOneResNet18DN.

To compare the performance of the standard model ResNet18, the VOneResNet18, and our new model, the VOneResNet18DN, we used the Tiny ImageNet validation dataset to evaluate their accuracy, and the common image corruptions dataset scaled to Tiny ImageNet [14] to evaluate their robustness. This last dataset contains 75 perturbations: 15 different corruption types each at five levels of severity, which can be grouped into four categories (noise, blur, weather, and digital). To evaluate in detail whether the current implementation of the divisive normalization model makes the V1 stage more aligned to primate V1, we used a set of benchmarks that evaluate model to brain alignment using single neuron response property distribution similarity [15, 35].

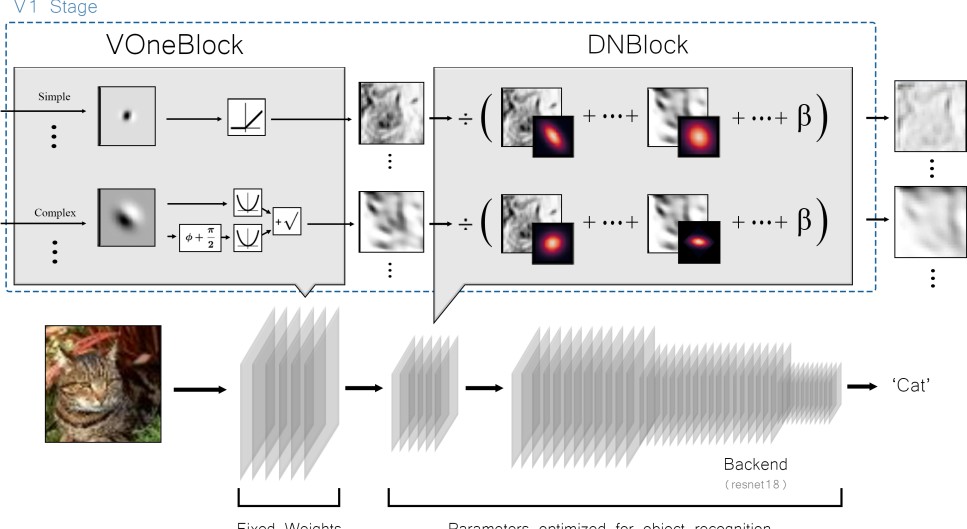

Figure 1: **Overview of the VOneNetDN model family with divisive normalization.** The VOneNet model family was expanded by introducing a divisive normalization model, the DNBlock, after the VOneBlock. The VOneBlock contains a convolutional layer (Gabor filter bank with weights constrained by empirical data) and a nonlinear layer (simple and complex cell nonlinearities). The output of the VOneBlock is then propagated through the DNBlock where each channel is normalized by a weighted sum of the activity in all the channels. Each channel has a distinct set of normalizing 2D Gaussian filters which are used to convolve all the VOneBlock channels. The VOneBlock together with the DNBlock form the V1 stage. The rest of the model back-end is a standard CNN architecture.

## 3 Results

### 3.1 Divisive normalization improves robustness to common image corruptions

When compared to the standard model (ResNet18), the VOneResNet18 (no divisive normalization) shows improved robustness to all common image corruptions categories (overall relative gain of 11% for top-1 accuracy). This, however, comes at a cost of a slightly lower performance on clean images (relative accuracy of 95%). The new model with divisive normalization, on the other hand, reaches virtually the same accuracy of the standard model (relative accuracy of 99%) and improves on all corruption categories by an even larger amount (relative gain of 18%, Figures 2 and A.2). Compared to the model without divisive normalization, the VOneResNet18DN is better on clean images and all corruption categories except blur, with the greatest individual improvements being for the fog and contrast corruptions (relative gains of 28% and 93%, Table A.3 and Figure A.2).

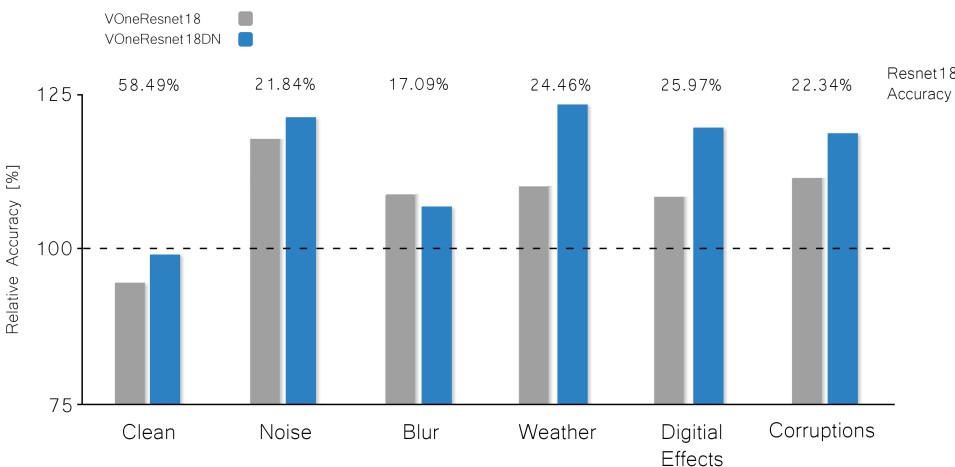

Figure 2: **Divisive normalization improves robustness to common corruptions.** Relative accuracy (top-1, normalized by ResNet18) on clean images, corruptions categories, and overall corruptions for the VOneResNet18 (gray), and the VOneResNet18DN (blue) (see Table A.3 for absolute accuracies).

## 3.2 Divisive normalization improves alignment to some but not all V1 response properties

For response properties in which divisive normalization mechanisms have been implicated, such as those related to contrast response function and surround modulation [23], the VOneResNet18DN obtained considerable gains over the model without divisive normalization (relative improvements of 36% and 148%, Figure 3 and Tables A.1 and A.2). Inspection of individual neurons contrast and size tuning curves in the model with divisive normalization revealed that similarly to biological neurons, these exhibited non-linear saturating responses with increasing contrast, as well as decreased responses for large stimuli (surround suppression). While the alignment to primate V1 improved on average with the introduction of the divisive normalization layer (relative gain of 10%), the benchmark scores for some response properties degraded (receptive field size, texture modulation, and response magnitude), suggesting that the current implementation of divisive normalization can still be further improved (Figure 3 and Tables A.1 and A.2).

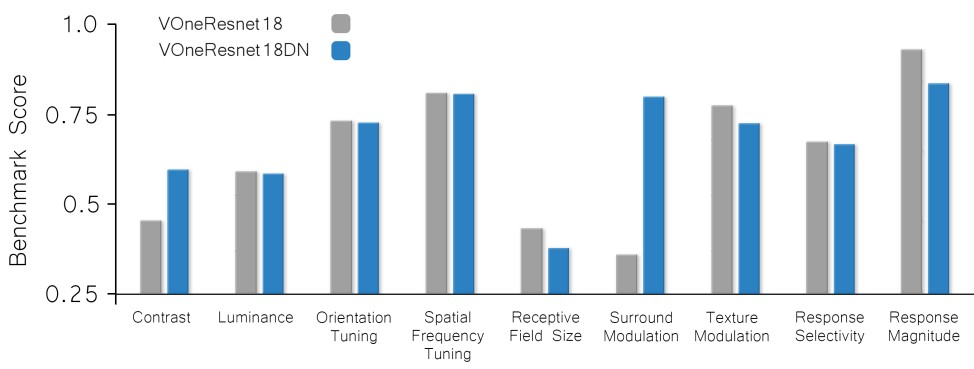

Figure 3: **Divisive normalization improves alignment to primate V1 for contrast and surround modulation properties.** Scores on set of V1 benchmarks that measure model alignment at the level of single-neuron response properties for VOneResNet18 (gray) and VOneResNet18DN (blue) (see Tables A.1 and A.2 for detailed results).

# 4 Discussion

Building models that more robustly generalize to image distributions unseen during training is a major problem in computer vision [13, 14]. State-of-the-art approaches to increase generalization and robustness of models typically rely on data augmentation and regularization procedures during training. Here, we took inspiration in neurobiology and introduced divisive normalization, a computational model widely used in neuroscience to explain multiple phenomena in low-level vision. Divisive normalization computations in V1 and other early vision areas are often credited for the ability of primate vision to process such a wide dynamic range of luminance and contrast in naturally occurring stimuli. Our new model containing divisive normalization improved robustness to common image corruptions when compared to a standard CNN model, and to a biologically-inspired model without the divisive normalization layer. In addition, it also improves in alignment to V1 according to an extensive set of brain benchmarks. While our model still falls short of completely addressing the problem of robustness and generalization, this work demonstrates that combining neuroscientific models with standard CNNs can lead to gains in computer vision applications.

Even though the VOneResNet18DN, was not trained with any data augmentation, it achieved considerably better performance in classifying corrupted images when compared to both the standard model, ResNet18, and the model without normalization, VOneResNet18. Corruption types related to contrast and weather perturbations corresponded to the largest performance gains. Since these tend to affect the luminance and contrast of the images either globally (contrast and brightness) or locally (fog and frost), the divisive normalization layer is likely to compensate for some of these perturbations in order to improve model accuracy. In addition, despite having a fixed-weight Gabor Filter Bank as its first convolutional layer, which usually degrades performance for clean images [18, 36], the model achieved virtually the same clean accuracy as the standard model, suggesting that a divisive normalization may compensate for some of the performance lost due to a more limited representation after the first convolutional stage.

It is known that optimizing for performance in a computer vision task, leads to improved alignment of standard CNNs to the the ventral stream [4] (though this result weakens for high-performing models [7]). Here we extend this result to optimizing a model containing a trainable divisive normalization layer for object classification and measuring its alignment to V1 using an extensive set of benchmarks [15]. Interestingly, the model with divisive normalization greatly improved its alignment to V1 for contrast and surround modulation response properties, which are thought to result from recurrent connectivity within V1. The spatial filters of the divisive normalization layer are likely capturing this type of processing. Unfortunately, the improvement of alignment to V1 was limited to these benchmarks, since for the remaining the scores remained unchanged or even slightly degraded.

While the model described in this study represents a relevant development in bridging neuroscience and computer vision models, the improvements obtained are limited and far from perfect. Despite the large improvements in performance for the common image corruptions, performance of the model is still low for high severity perturbations (Figure A.2), specially those of the blur category, the only category that the divisive normalization layer led to a reduction in accuracy. Future work combining the divisive normalization layer with other training-based approaches, such as using different types of data augmentation, may potentially lead to further improvements in model robustness and generalization. As for the brain-benchmarks, the divisive normalization layer did not improve alignment to V1 for any of the benchmarks categories other than contrast and surround modulation. That is not entirely surprising as the divisive normalization was added to the VOneBlock, a model component that had been designed separately. More extensive analyses dissecting neuronal responses in the model may provide insights on how to change both the components of the V1 stage, the VOneBlock and the DNBlock, in order to improve their alignment to primate V1.

This study builds on a growing literature exploring the benefits of combining divisive normalization models with CNNs to build hierarchical models of the ventral stream and object recognition. In the future, it will be interesting to study the benefits of adding divisive normalization layers in multiple stages of the model. In addition, here, we limited the model to brain comparisons to primate V1. It remains to be seen whether adding a divisive normalization layer in early model layers, lead to more brain-like representations in downstream layers. While TinyImageNet is already a considerably more difficult problem that CIFAR, expanding the current models to the ImageNet dataset is a critical next step towards building a model that can potentially be used for real-world applications and can better emulate visual processing in the primate ventral stream.

## Acknowledgments and Disclosure of Funding

This work was supported by the Army Research Office Grant W911NF2110192 (A.C , R.V., A.B. , and L.U.), the MIT MLK Visiting Professor Fellowship (L.U.), the PhRMA Foundation Postdoctoral Fellowship in Informatics (T.M.), the Semiconductor Research Corporation (SRC) and DARPA (J.J.D.), Office of Naval Research grant MURI-114407 (J.J.D.), the Simons Foundation grant SCGB-542965 (J.J.D.), the MIT-IBM Watson AI Lab grant W1771646 (J.J.D.).

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

# Supplementary Material

## A    Supplementary Material

### A.1    Tiny ImageNet

We used the Tiny ImageNet dataset for model training and evaluating model clean accuracy [34]. Tiny ImageNet training dataset contains 100.000 colored images of 200 classes (500 for each class) at a resolution of 64×64 pixels. In addition, there are 50 additional images per class for validation. Tiny ImageNet is publicly available at `https://www.kaggle.com/c/tiny-imagenet`.

### A.2    Common Image Corruptions

Tiny ImageNet-C is a dataset that contains the validation images from Tiny ImageNet under multiple forms of corruption [37]. Tiny ImageNet-C contains 15 different types of corruption at 5 levels of severity. These corruptions are Gaussian noise, shot noise, impulse noise, defocus blur, glass blur, motion blur. zoom blur, snow. frost, fog, brightness. contrast, elastic transform, pixelation, and JPEG compression. See Figure A.1 for examples.

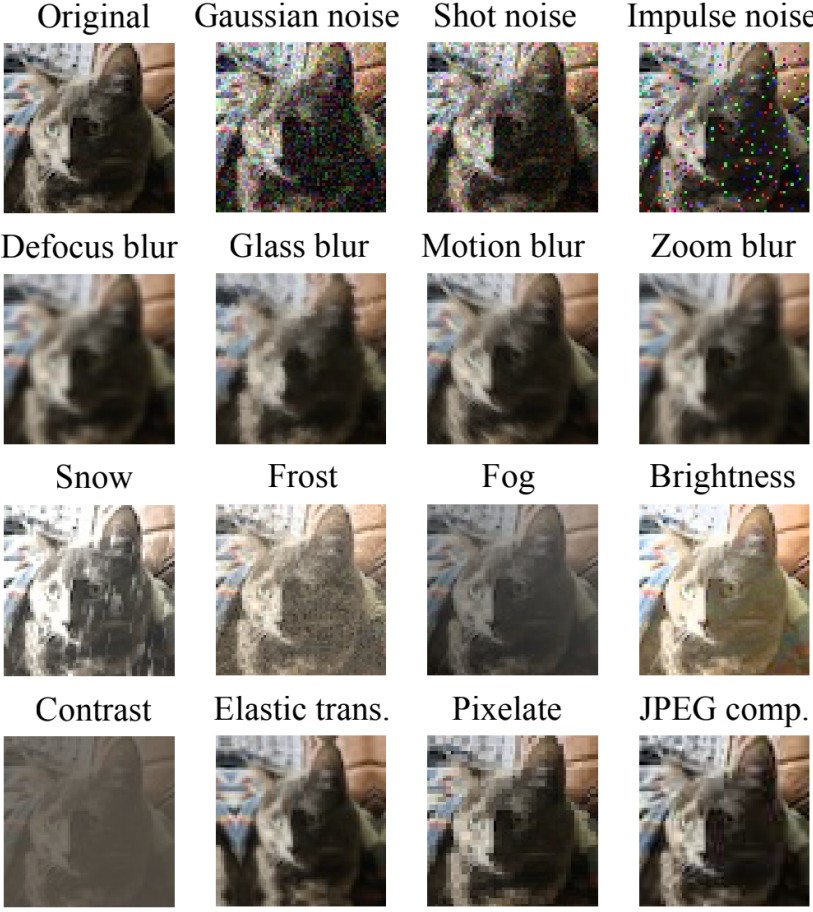

Figure A.1: **Common image corruptions at Tiny ImageNet resolution.** All 15 types of common image corruptions with a severity of 3.

### A.3 Models

The following VOneNet and ResNet18 descriptions in the present paper are derived from Baidya et al. 2021 [36].

#### A.3.1 VOneNet

VOneNet [18] is a convolutional neural network with a front-end (VOneBlock) simulating the primary visual cortex (V1). In the present study, we used a modified ResNet18 architecture as the architecture back-end. The VOneBlock contains a fixed weight Gabor filter bank (GFB) [38] used to mimic the receptive fields found in V1. Each Gabor filter is constructed using parameters generated from empirically observed distributions in preferred orientation, spatial frequency, and size of receptive fields [39], [40], [41]. The Gabor filters are generated to simulate 256 simple and 256 complex cell receptive fields. Several changes were made to the VOneNet for compatibility with Tiny ImageNet as the original model was built for ImageNet. Importantly, the field-of-view of the model corresponding to the 64×64px inputs was adjusted to 2deg, giving the model a resolution of 32 pixels per degree (ppd). Further details on these changes can be found in [36]. Code for the VOneNet is publicly available at https://github.com/dicarlolab/ vonenet under GNU General Public License v3.0.

#### A.3.2 ResNet18

We used a modified Resnet18 architecture as both a base model and as a back-end for VOneNet. The Resnet18 architecture was modified such that the stride of the first convolutional layer was changed from two to one and the first maxpool layer was kept at a stride of two. This results in a combined stride of two in the first block, which is the same as the VOneBlock. Baidya et al. [36] found that this modification leads to a significant improvement in accuracy, with the modified ResNet18 achieving an accuracy of 58.93% when trained and evaluated on Tiny ImageNet in comparison to 50.45% prior to the modification.

### A.4 Adding Divisive Normalization to VOneNet

The implementation of divisive normalization in this study was based on its generalized form which features in Burg et al. [32]. Neuronal responses of a neuron $l$ to stimulus $x$ $(y_l(x))$ are normalized by the a factor that depends on the responses of each neuron $k$ in a pool of $K$ neurons ($y_k(x)$ for $k \in K$), according to the following equation:

$$z_l(x) = \frac{y_l^{n_l}(x)}{\sigma_l^{n_l} + \Sigma_{k \in K} P_{kl} \cdot y_k^{n_k}(x)} \quad (1)$$

where $z_l$ represents the normalized response of neuron $l$, $\sigma_l$ is a semi-saturation constant, $n_l$ represents a learned parameter relative to neuron $l$ to exponentiate $y_l(x)$ and $\sigma_l$, and $P_{kl}$ represents the normalization weights of neuron $k$ onto neuron $l$.

We adapted this formulation to create the DNBlock to serve as a divisive normalization module directly following VOneBlock (see Figure **??**). The output of the VOneBlock consists of 512 32×32 response channels per image. For an image $x$ and a Gabor filter $l$, the response channel is $Z_l(x)$. To divisively normalize the response channel $Z_l(x)$ with respect to each response channel $Z_k(x)$ in a pool of $K$ response channels, we used the following version of the previous equation 1:

$$\bar{Z}_l(x) = \frac{Z_l(x)}{\beta + \sum_{k \in K} \alpha_{kl} \cdot Z_k(x)} \quad (2)$$

where $\bar{Z}_l(x)$ represents the divisively normalized response channel $Z_l(x)$, $\beta$ is a bias term and $\alpha_{kl}$ represents the normalization weights of response channel $Z_k(x)$ onto response channel $Z_l(x)$. The exponential term $n$ has not been included as, in training, caused significant instability. In [32], the normalization weights $P_k l$ were learned without constraint and covered 0.5 degrees of the visual field, which leads to only local interactions being captured. In our modified implementation, the normalization weights $\alpha_{kl}$ were constrained to a two-dimensional Gaussian kernel, greatly reducing the number of parameters to be learned, and covered about two degrees of the visual field, thus capturing more distal interactions. The equation for building the Gaussian kernel is the following:

$$\alpha_{kl} = G(k, l | \theta, v, w, \rho, \sigma, A) = \frac{A}{2\pi\rho\sigma}\exp\left(-\frac{1}{2}\left[\frac{(x_{rot} - v)^2}{\rho^2} + \frac{(y_{rot} - w)^2}{\sigma^2}\right]\right) \quad (3)$$

where

$$x_{rot} = x cos(\theta) + y sin(\theta)$$
$$y_{rot} = -x sin(\theta) + y cos(\theta)$$

Each parameter of this kernel is optimized during training. Each kernel is unique to each combination of channel being normalized and channel normalizing it, resulting in $512^2 = 262,144$ total Gaussian kernels.

## A.5 Training

Extensive details on image preprocessing, loss functions, and optimization, can be found in Baidya et al. [36]. In summary, image preprocessing consisted of randomly scaling each image by a factor of 1-1.2 and randomly rotating each image between 30 and -30 degrees. Images were shifted between -5% and 5% of the images total height and width the vertical and horizontal directions respectively. Images were normalized by subtraction and division by [0.5, 0.5, 0.5]. The model was optimized using Stochastic Gradient Descent with a momentum of 0.9, a weight decay of 0.0005, and an initial learning rate of 0.1. The learning rate was reduced by a factor of 10 whenever 5 training epochs passed simultaneously with no improvement in accuracy. Models were trained with an image batch size of 128 over 60 epochs. Weight decay was disabled for all parameters in the DNBlock.

## A.6 Brain-Score Benchmarks

The Brain-Score platform [35, 7] offers several benchmarks to test the biological accuracy. These benchmarks may be found at `https://www.brain-score.org/`. To compare the models with and without divisive normalization, we used benchmarks that measure the alignment of model to V1 at the level of single-neuron response properties. These benchmarks measure the similarity between the distributions of single neuron response properties in the model and V1. Detailed description of 22 of these benchmarks can be found in Marques, et al. 2021 [15]. These are grouped in seven categories: orientation tuning, spatial frequency tuning, receptive field size, surround modulation, texture modulation, response selectivity, and response magnitude. Here, we also included eight new benchmarks from contrast and luminance response property categories described in a study under review (will be updated to include citation).

### A.6.1 Contrast

For these benchmarks, achromatic sinusoidal gratings of varying contrast are presented at the preferred orientation and spatial frequency of each model neuron. Then, for each neuron, its responses to different contrasts are fit using the hyperbolic function: $R = R_{max}\frac{c^n}{c^n + c_{50}^n}$

Distributions of four response properties were used to compare the fits of brain and model neurons; the number of standard contrast responses, the maximum response $R_{max}$, the semisaturation constant($c_{50}^n$), and the exponent $n$ of the function.

### A.6.2 Luminance

For luminance benchmarks, uniform stimuli varying from 0.1 cd/m$^2$ to 100 cd/m$^2$ in seven steps on a logarithmic scale were presented. For each neuron, luminance tuning curves were calculated and the responses to dark (below 3 cd/m$^2$) and bright (above 3 cd/m$^2$) stimuli were fitted logarithmically.

Four response properties were used in these benchmarks: the number of surface luminance response neurons, the slope of the firing rate versus log(luminance) for dark and bright stimuli and the normalized difference between the two slopes.

## A.7 Detailed Results

**Contrast**

| Benchmark | VOneResnet18 Performance | VOneResnet18 DN Performance |
|---|---|---|
| Standard Neuron | **1.000** | 0.996 |
| Maximum Response | 0.161 | **0.689** |
| Semisaturation Constant | 0.192 | **0.345** |
| Exponent | 0.677 | **0.736** |

**Luminance**

| Benchmark | VOneResnet18 Performance | VOneResnet18 DN Performance |
|---|---|---|
| Surface Responsive | 0.762 | **0.852** |
| Dark Slope | **0.846** | 0.756 |
| Bright Slope | **0.385** | 0.367 |
| Delta Slope Norm | **0.746** | 0.706 |

**Orientation Tuning**

| Benchmark | VOneResnet18 Performance | VOneResnet18 DN Performance |
|---|---|---|
| Circular Variance | **0.766** | 0.731 |
| Or. Bandwidth | 0.923 | **0.956** |
| Orth. Pref. Ratio | **0.725** | 0.675 |
| OR. Selective | **1.000** | 0.994 |
| CV Bandwidth Ratio | 0.770 | **0.793** |
| Opr. CV Diff | **0.886** | 0.883 |
| Preferred Orientation | 0.991 | **0.994** |

**Spatial Frequency Tuning**

| Benchmark | VOneResnet18 Performance | VOneResnet18 DN Performance |
|---|---|---|
| Peak SF. | **0.981** | 0.929 |
| SF. Selectivity | 0.981 | **0.984** |
| SF. Bandwidth | 0.937 | **0.977** |

**Receptive Field Size**

| Benchmark | VOneResnet18 Performance | VOneResnet18 DN Performance |
|---|---|---|
| Grating Summation Field | **0.589** | 0.426 |
| Surround Diameter | 0.370 | **0.392** |

Table A.1: **Single neuron response property V1 benchmarks (Contrast, Luminance, Orientation Tuning, Spatial Frequency Tuning, and Receptive Field Size categories)**.

| Surround Modulation | | |
|---|---|---|
| Benchmark | VOneResnet18 Performance | VOneResnet18 DN Performance |
| Surround Suppression Index | 0.385 | **0.954** |

| Texture Modulation | | |
|---|---|---|
| Benchmark | VOneResnet18 Performance | VOneResnet18 DN Performance |
| Texture Modulation Index | **0.898** | 0.836 |
| Absolute Texture Modulation Index | **0.944** | 0.880 |

| Response Selectivity | | |
|---|---|---|
| Benchmark | VOneResnet18 Performance | VOneResnet18 DN Performance |
| Texture Selectivity | 0.795 | **0.958** |
| Texture Sparseness | **0.927** | 0.783 |
| Texture Variance Ratio | **0.719** | 0.674 |
| Modulation Ratio | **0.723** | 0.718 |

| Response Magnitude | | |
|---|---|---|
| Benchmark | VOneResnet18 Performance | VOneResnet18 DN Performance |
| Max DC | 0.838 | **0.976** |
| Max Texture | **0.914** | 0.688 |
| Max Noise | **0.923** | 0.725 |

Table A.2: **Single neuron response property V1 benchmarks (Surround Modulation, Texture Modulation, Response Selectivity, and Response Magnitude categories).**

| | | Noise | | | Blur | | | |
|---|---|---|---|---|---|---|---|---|
| Model | Clean [%] | Gaussian [%] | Shot [%] | Impulse [%] | Defocus [%] | Glass [%] | Motion [%] | Zoom [%] |
| ResNet18 | **58.5** | 20.0 | 23.1 | 22.4 | 13.9 | 18.9 | 19.6 | 15.9 |
| VOneResNet18 | 55.3 | 24.3 | 28.7 | 24.4 | **15.6** | **19.7** | **21.6** | **17.2** |
| VOneResNet18DN | 57.8 | **25.2** | **29.2** | **24.6** | 15.0 | **19.7** | 21.4 | 16.6 |

| | Weather | | | | Digital | | | |
|---|---|---|---|---|---|---|---|---|
| Model | Snow [%] | Frost [%] | Fog [%] | Bright. [%] | Contrast [%] | Elastic [%] | Pixelate [%] | JPEG [%] |
| ResNet18 | 24.1 | 25.2 | 21.6 | 27.0 | 9.8 | 24.3 | 37.8 | 32.0 |
| VOneResNet18 | 27.5 | 27.6 | 22.9 | 28.9 | 9.5 | **28.5** | 36.6 | 37.3 |
| VOneResNet18DN | **29.0** | **30.1** | **29.2** | **31.6** | **18.3** | **28.5** | **38.5** | **38.2** |

Table A.3: **Absolute accuracies (top-1) of Resnet18, standard VOneResnet18 and VOneResnet18DN (averaged over perturbation severities).**

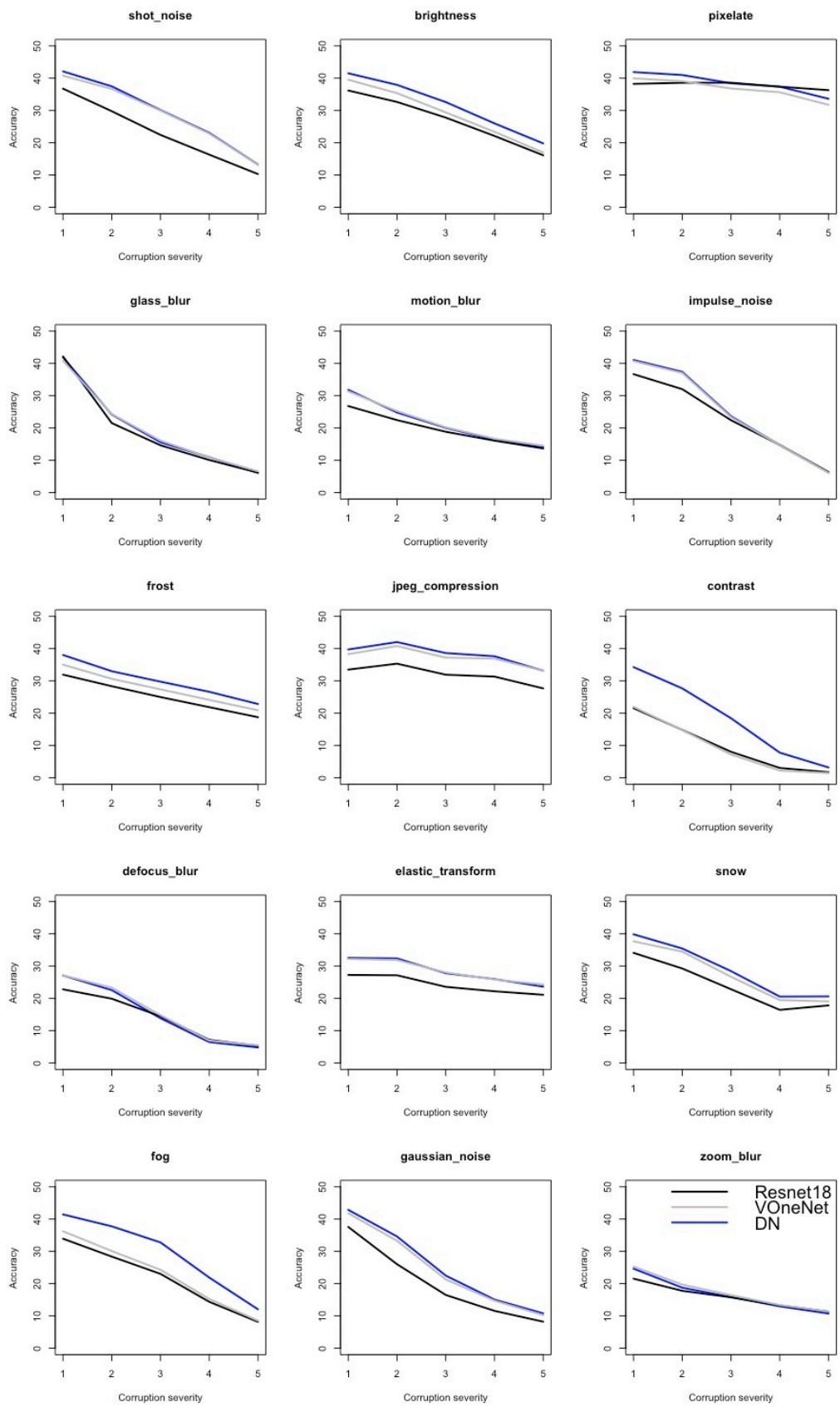

Figure A.2: **Absolute accuracies (top-1) on common corruptions for ResNet18, VOneResNet18, and VOneResNet18DN.** All 15 types of common corruptions at all perturbation severity levels.

