# OpenReview forum: "Implementing Divisive Normalization in CNNs Improves Robustness to Common Image Corruptions"
_NeurIPS.cc/2022/Workshop/SVRHM — SVRHM Poster_

### Official Review · Reviewer_vLXG · 2022-10-12
**Good paper, combining two established bio-inspired approaches to enhance the robustness of CNNs.**

**Rating:** 7
**Confidence:** 4

**Review:**

The authors propose appending a divisive normalization (DN) layer to the VOneBlock, resulting in a novel VOneNetDN based on the ResNet-18. Both the layer and block are inspired by biological vision, and the authors show an increase in robustness against typical image corruptions when comparing the novel architecture to the ResNet-18. In addition, the new approach increased the lower clean accuracy observed with the original VOneNet, almost matching the clean accuracy of the ResNet-18.
Moreover, the authors measure a better V1 alignment of the VOneBlockDN.

To my mind, this is a solid paper that is clearly written, showing good results with convincing and well-discussed experiments that are –thanks to an extensive appendix – also reproducible.

The authors may improve the paper further by commenting on why they expected that DN improves CNN robustness. Maybe they can offer theoretical considerations as in [Paiton et al.] and [Grüning et al.] (see references below).

Considering the weaknesses of the paper, I find the use of relative accuracy metrics misleading: arguably, 'with the VOneNetDN, we obtained 99% relative clean accuracy of the ResNet-18' sounds better than 'with 57.8% the VOneNetDN's clean accuracy is lower than the ResNet-18's accuracy (58.5%)'. However, with the second metric, a comparison to other approaches is far easier. Moreover, the VOneNetDN's increase of absolute clean accuracy over the VOneNet with only slightly changing the architecture is a result worth publishing anyway.
Thus, I recommend moving Table A.3 from the Appendix to the main text, moving Figure 2 to the Appendix instead, and referring to the absolute accuracy and relative accuracy metrics in the results and discussion sections.

The authors should consider mentioning other methods of using bio-inspired approaches to increase the robustness of neural networks, for example:
Paiton et al.: Selectivity and robustness of sparse coding networks (Journal of Vision 2020),
Grüning et al.: FP-nets as novel deep networks inspired by vision (Journal of Vision 2022).

When referencing DN, the authors should cite the work by Simoncelli and his group.

Finally, there is a certain lack of novelty: the presented VOneNetDN is only a slight adaptation of the VOneNet, using the already well-researched DN layer. Hence, by assessing the paper as good but not exceptional, I give a rating of 7.

---

### Official Review · Reviewer_ge6H · 2022-10-13
**Adding a divisive normalization block to a VOneNet increases robustness and V1 alignment – solid well written paper – more thorough and precise presentation of related work would be favorable**

**Rating:** 7
**Confidence:** 3

**Review:**

In this paper, the authors extend a bio-inspired hybrid CNN (VOneNet; a combination of a biologically constrained model of the primate V1 as front-end, and a standard CNN back-end) with a divisive normalization (DN) block. They show that doing so increases robustness to image perturbations and alignment to primate V1 wrt nonlinear contrast responses and surround modulation (as compared to the same model without a DN block).

The results, which show that a DN block in combination with a VOneNet improves robustness against image perturbations, seem novel and relevant to the field. However, I have some concerns regarding the presentation of related work as well as regarding the relevance and novelty of the presented results regarding the alignment to primate V1.

Overall, I think the paper is straightforward, well written and provides some relevant insights regarding the mechanisms that potentially explain the remarkable robustness of biological visual systems against image perturbations. Thus, I suggest acceptance but would encourage the authors to consider the following points.

- The authors use an already existing model (VOneNet) and combine it with a well-known neuroscientific model/idea (DN) which is known to explain surround suppression and nonlinear contrast response characteristics and show that the resulting model can better explain surround suppression and nonlinear contrast response characteristics than a model without DN. I may overlook some important aspects here, but this particular finding strikes me as rather trivial. It could be made clearer what the novel aspects of the presented work are, and how they differ from other related work.

- I think it would be beneficial to elaborate more on the implementations of DN in the presented model. How is it different from other implementations of DN in CNNs? How is it that other studies [1] find only a benefit of DN in shallow CNNs? How does the implementation here differ from the implementation in [2] (which btw I think is a relevant paper wrt to DN and CNNs and should be cited here). Why is the DN block implemented at this particular stage in the CNN? It is assumed that DN also plays a crucial role in the retina and in later stages of visual processing [3]. Why not implement multiple DN blocks at different stages in the CNN? These are questions that should be addressed in the paper.

- The elaboration on other work investigating DN in combination with CNNs should not only be more extensive but also more precise: In lines 58-59, the authors state that “In [4], a CNN with divisive normalization improved neuronal predictivity in the macaque V1.” This can be understood such that in [4] a very similar approach has been applied which led to very similar results as presented here. However, this seems not the case, the authors in [4] added DN to a model which only features one convolutional layer but no deep architecture. In other words, it would be valuable to be more precise regarding the differences between the presented work and already conducted studies. Otherwise, the reader might gain a wrong impression regarding the novelty of the presented work.

- Further, I miss further model comparisons, which would allow for a better assessment of the implication of the presented work. For example, it would be interesting to see how a model only featuring the DN block, but no VOne block is performing. Comparisons with models featuring a DN block at different stages or multiple DN blocks would also be valuable. Further, a comparison with other implementations of DN (e.g., such as in [2]), would be interesting.

- It also would be exciting to evaluate the model on texture-shape cue-conflict images. This would allow checking whether adding a DN block improves robustness by preventing shortcut learning and/or increasing shape bias (compare [5]).

Minor comments:

- Line 19: You mention that “… CNNs have also been *relatively* successful in modelling neuronal responses… “(emphasis added). Relative to what? Models which do not feature convolutions? Models not featuring a deep architecture? Either write they are successful, or they are successful relative to something, which must be specified.

- Figure 3: I would suggest including the performance of the vanilla ResNet18 in the plot.

- Line 121: You write: “*Even though the model with divisive normalization, the VOneResNet18DN*, was not trained with any data augmentation…”(emphasis added). Be more concise, just write “*The VOneResNet18DN…*” – by now, everybody is aware that this refers to your model with divisive normalization.

- Line 368: The hyperlink to the Figure seems broken.


[1] Pan, X., Giraldo, L. G. S., Kartal, E., & Schwartz, O. (2021). Brain-inspired weighted normalization for CNN image classification. *bioRxiv*.

[2] Ren, M., Liao, R., Urtasun, R., Sinz, F. H., & Zemel, R. S. (2016). Normalizing the normalizers: Comparing and extending network normalization schemes. *arXiv preprint arXiv:1611.04520*.

[3] Carandini, M., & Heeger, D. J. (2012). Normalization as a canonical neural computation. *Nature Reviews Neuroscience*, 13(1), 51-62.

[4] Burg, M. F., Cadena, S. A., Denfield, G. H., Walker, E. Y., Tolias, A. S., Bethge, M., & Ecker, A. S. (2021). Learning divisive normalization in primary visual cortex. *PLOS Computational Biology*, 17(6), e1009028

[5] Geirhos, R., Rubisch, P., Michaelis, C., Bethge, M., Wichmann, F. A., & Brendel, W. (2018, September). ImageNet-trained CNNs are biased towards texture; increasing shape bias improves accuracy and robustness. In *International Conference on Learning Representations*.

---

### Official Review · Reviewer_LDgi · 2022-10-14
**Interesting application of divisive normalization to image recognition**

**Rating:** 7
**Confidence:** 5

**Review:**

### Summary
The authors present the application of the well-studied divisive normalization operation to deep convolutional networks (with the VOneNet block) trained to perform image recognition. The authors show that without a considerable drop in clean performance, the robustness of networks with DN to common image corruptions is enhanced. They also present mixed results on how well the networks w/ and w/o DN capture a set of characteristic properties of V1 neurons. I find this work to be relevant to SVRHM and think it will trigger interesting discussions at the workshop. Hence I recommend accepting this paper.

### Pros
- Incorporating insights from biological vision into deep learning models is a promising direction, especially in making the latter robust to common image corruptions. This submission demonstrates that biologically inspired divisive normalization can enhance perceptual robustness and potentially slightly improve representational similarity to biological vision.
- This particular implementation of DN is quite efficient as it introduces only a few trainable extra parameters (mean and variance of the Gaussian DN kernels)
- It's interesting that VOneNet affects clean image performance but the addition of DN diminishes drop in accuracy.

### Room for improvement
- As the authors rightly point out, there are still several V1 properties that the model doesn't yet capture. This gives room for improving the model aimed at increasing representational similarity to biological V1 neurons. Have the authors explored relaxing the Gaussian shape constraint on DN kernels and making them fully trainable?


The authors may also be interested in looking at more recent work on applying DN to convolutional networks that are relevant to their submission here:
1. Miller, M., Chung, S., & Miller, K. D. (2021, September). Divisive Feature Normalization Improves Image Recognition Performance in AlexNet. In International Conference on Learning Representations.
2. Veerabadran, V., Raina, R., & de Sa, V. R. (2021, October). Bio-inspired learnable divisive normalization for ANNs. In SVRHM 2021 Workshop@ NeurIPS.

---

### Official Review · Reviewer_cuDx · 2022-10-16
**An interesting investigation of divisive normalization in the context of image classification robustness.**

**Rating:** 7
**Confidence:** 3

**Review:**

The submission augments a class of standard convolutional neural networks with divisive normalization, demonstrating improvement in robustness to common image corruptions, and quantitative fit to imaging data from primates. The investigation is detailed and will be of interest to researchers at the workshop. A missing set of baselines is previously-introduced methods using divisive normalization in computer vision, including those already cited in the Related Work section—instead, the submission compares the implemented modification to a standard model. Nevertheless, the findings are well-placed in the context of broader discussion that are central themes at the workshop, including human-machine comparisions and robustness in human and machine vision.